# Lifestyle Interventions to Improve Glycemic Control in Adults with Type 2 Diabetes Living in Low-and-Middle Income Countries: A Systematic Review and Meta-Analysis of Randomized Controlled Trials (RCTs)

**DOI:** 10.3390/ijerph18126273

**Published:** 2021-06-10

**Authors:** Grainne O’Donoghue, Cliona O’Sullivan, Isabelle Corridan, Jennifer Daly, Ronan Finn, Kathryn Melvin, Casey Peiris

**Affiliations:** 1School of Public Health, Physiotherapy and Sports Science, University College Dublin, D04 V1W8 Dublin, Ireland; cliona.osullivan@ucd.ie (C.O.); isabelle.corridan@ucdconnect.ie (I.C.); jennifer.daly@ucdconnect.ie (J.D.); ronan.finn29@gmail.com (R.F.); kathryn.melvin@ucdconnect.ie (K.M.); 2School of Allied Health, Human Services and Sport, La Trobe University, Melbourne, VIC 3086, Australia; C.Peiris@latrobe.edu.au

**Keywords:** lifestyle intervention, dietary advice, physical activity guidelines, self-management programs, type 2 diabetes, low-middle income countries, HBA1c, FBG, body mass index, systematic review, meta-analysis

## Abstract

Alongside glucose lowering therapy, clinical guidelines recommend lifestyle interventions as cornerstone in the care of people living with type 2 diabetes (T2DM). There is a specific need for an up-to-date review assessing the effectiveness of lifestyle interventions for people with T2DM living in low-and-middle income countries (MICs). Four electronic databases were searched for RCTs published between 1990 and 2020. T2DM, lifestyle interventions, LMICs and their synonyms were used as search terms. Data codebooks were developed and data were extracted. Narrative synthesis and meta-analysis were conducted using random effects models to calculate mean differences (MD) and standardized mean differences (SMD) and 95% confidence intervals (CI). Of 1284 articles identified, 30 RCTs (*n* = 16,670 participants) met the inclusion criteria. Pooled analysis revealed significant improvement in HBA1c (MD −0.63; CI: −0.86, −0.40), FBG (SMD −0.35; CI: −0.54, −0.16) and BMI (MD −0.5; CI: −0.8, −0.2). In terms of intervention characteristics, those that included promoted self-management using multiple education components (e.g., diet, physical activity, medication adherence, smoking cessation) and were delivered by healthcare professionals in a hospital/clinic setting were deemed most effective. However, when interpreting these results, it is important to consider that most included studies were evaluated as being of low quality and there was a significant amount of intervention characteristics heterogeneity. There is a need for further well-designed studies to inform the evidence base on which lifestyle interventions are most effective for glycemic control in adults with T2DM living in LMICs.

## 1. Introduction

Between 1980 and 2014, the global prevalence of diabetes nearly doubled from 4.7% to 8.5%, with most new cases in low-and-middle income countries (LMIC) [1]. South-East Asia has an estimated 96 million people living with diabetes (90% T2DM), a significantly higher prevalence than within the World Health Organization’s (WHO) European (*n* = 60 million) or American (*n* = 47 million) regions [2,3]. Furthermore, the WHO 2016 Global Report stated that, since 2012, the lower socioeconomic class of middle-income countries (MIC) had the highest mortality attributed to hyperglycemia and T2DM, which is detrimental to population health in LMICs [4]. It has significant economic consequences, threatening to reverse economic gains in developing countries, which will further limit resources, making them even less equipped to manage this diabetes epidemic [5].

In addition to glucose-lowering therapies, lifestyle management, including nutritional education, physical activity (PA), weight loss and self-efficacy education is recommended as a cornerstone in the care of a patient living with T2DM [6]. Healthcare and community-based lifestyle interventions have been shown to be both clinically and cost-effective ways of curbing the growth of diabetes [7]. A substantial number of systematic reviews that mainly include studies conducted in high income countries provide considerable evidence supporting lifestyle interventions as cost and clinically effective in the prevention [8,9] and management of T2DM by reducing body weight and improving glucose control [10,11,12].

There are a number of reviews exploring the evidence for the prevention [9,13] and lifestyle management [11,12,13] of T2DM in LMICs. Previous reviews of management interventions have focused on specific regions or continent sub-regions, such the Asian Western Pacific (AWP) [11] region and Southeast Asia [12], a specific type of intervention, for example, diabetes self-management education [11] or specific site of delivery [13]. A review published in 2016 [14] that included literature up until 2014 for both prevention and management trials in LMICs concluded that there was a lack of large experimental trials exploring diabetes control. Moreover, the quality of the documented studies was poor and, due to the heterogeneity in the reported interventions, the authors were not able to pool findings; hence, reported results are only based on descriptive data.

Therefore, the aim of this review was to synthesize the most up-to-date evidence in relation to lifestyle interventions that target glycemic control in adults with T2DM living in LMICs. Our specific objectives were:To explore core components of lifestyle interventions, specifically in relation to strategies and intervention characteristics for the management of T2DM (method, context of delivery, by whom, with what intensity and for how long).To complete a quality assessment on included randomized control trials (RCTs).To conduct a meta-analysis to establish the effectiveness of these interventions on improving glycemic control and anthropometry.To compile recommendations for future research on this topic.

## 2. Materials and Methods

### 2.1. Registration

This review was registered prospectively on the International prospective register of systematic reviews (PROSPERO 2020: CRD42020151938) and conducted in accordance with the Preferred Reporting Items for Systematic Reviews and Meta-analyses guidelines. [15].

### 2.2. Search Strategy

Four electronic databases (PubMed, CINAHL, Embase and CENTRAL) were searched. The search strategy was constructed around the PICOS tool: (P) Population: adults living in LMIC with type 2 diabetes; (I) Intervention: any intervention specifically targeting lifestyle behaviors; (C) Comparator: usual care, wait-list control, placebo; (O) Outcomes: glycemic control; and (S) Study Type: randomized controlled trials (RCTs). A complete list of the search terms is available in Appendix A. In addition to the databases, the reference lists of included articles were scanned for additional studies that met the inclusion criteria.

### 2.3. Inclusion Criteria

RCTs published in English and conducted in LMICs between January 1990 and October 2020 were included. Adults were defined as ≥18 years. T2DM was defined using one or more of the following criteria: fasting blood glucose (FBG) of ≥126 mg/dL (7.0 mmol/L) and/or 2-h FBG ≥ 200 mg/dL (11.1 mmol/L) and/or glycosylated hemoglobin (HbA1c) ≥ 6.5% [16]. LMICs were classified using the World Bank Classifications from 2019 [17]. A lifestyle intervention was any intervention that included a focus on increasing physical activity and/or changing dietary habits.

### 2.4. Outcome Measures

The primary outcome was change in glycemic control, as measured by glycated hemoglobin (HBA1c). Secondary outcomes were fasting blood glucose (FBG), other measures of metabolic risk (high density lipoproteins (HDL), low density lipoproteins (LDL), systolic and diastolic blood pressure (SBP and DBP), triglycerides (TG)), anthropometry (body weight, body mass index (BMI), waist circumference and percentage body fat) and quality of life.

### 2.5. Selection Process

All studies were imported into EndNote (Version 9X) and de-duplicated. Two authors independently screened titles, and subsequently abstracts, for potential inclusion (RF, IC). Following a review for accuracy, full texts were retrieved and independently assessed for inclusion in the review (JD, KM). Any disagreement over the eligibility was resolved through discussion with an additional reviewer (GO’D).

### 2.6. Data Extraction

Microsoft excel was used to develop a comprehensive codebook for data extraction. The following information was recorded: study details, participant demographics, intervention components (coded according to the Template for Intervention Description and Replication Checklist (TIDieR), comparator information, primary and secondary outcomes at baseline and follow-up and quality assessment material. The same authors involved in the screening process extracted data into the codebook.

### 2.7. Quality Assessment

To assess the risk of bias, the Effective Public Health Practice Project quality assessment tool was used [18]. All articles were assessed by two independent reviewers (RF and IC) and discrepancies were resolved with an additional reviewer (CO’S).

### 2.8. Data Synthesis

Firstly, the included trials were qualitatively described. The narrative synthesis included sample demographics, primary and secondary outcomes, intervention components and research quality. Quantitatively, meta-analyses were conducted in RevMan (version 5) for the primary outcome (HBA1c) and secondary outcomes of interest (FBG, BMI). Post-intervention means and standard deviations were pooled using random-effects models and inverse variance methods to calculate mean differences (MD), where outcomes were homogenous, or standardized mean differences (SMD) where outcomes were reported in different units and at 95% confidence intervals (CI). Where outcomes were measured at multiple timepoints, the post-intervention data were chosen to be pooled a priori as not all trials had follow-up measurements. For the primary outcome of HbA1c, measurements were only pooled if they were taken at least 12 weeks after baseline testing, as HbA1c is a measure of long-term glycaemic control over 2 to 3 months. Statistical heterogeneity was assessed using the I^2^ statistic, where I^2^ values >50% were considered to represent substantial heterogeneity [19]. Sub-group analyses were conducted to explore whether intervention type, delivery or setting, were observed to impact results. Intervention type was defined as either self-management education, consisting of a multifactorial program that focused on providing education around multiple risk factors associated with T2DM, or a structured nutrition/exercise intervention, where the focus was on prescribed or supervised individualized programming. Intervention delivery was categorized as being delivered by a healthcare Professional (HCP) or healthcare multidisciplinary team, or as being delivered by peers. Additionally, setting was based on whether the intervention was delivered in a hospital, community or home environment.

### 2.9. Assessment of Publication Bias

Where we included at least 10 studies in a meta-analysis, we generated funnel plots of effect estimates against their standard errors (on a reversed scale) using RevMan software. We assessed the potential risk of publication bias through visual analysis of the funnel plots. Roughly symmetrical funnel plots indicated a low risk of publication bias and asymmetrical plots a high risk. As this is a subjective judgement and asymmetry might also arise from other sources, we further attempted to avoid publication bias by not including multiple publications referring to the same data.

### 2.10. Grading of the Quality of Evidence

To evaluate the quality of the evidence in the subgroup analysis, the Grading of Recommendations, Assessment, Development and Evaluation (GRADE) approach was used. As all included trials were randomized controlled trials, the quality rating starts at “high”. All studies were allocated five points initially and were downgraded by one point, respectively, if there was evidence of: 1. risk of bias (40% or more of the trials had a global rating of week on the EPHPP tool); 2. unexplained heterogeneity (I^2^ > 50%); 3. indirectness in outcomes or population; 4. imprecision in results (wide 95% CI >  0.8) for SMD; or 5. publication bias (visual inspection of funnel plots when there were at least 10 trials in the meta-analysis). The quality of these data was then graded as follows: 1 = very low; 2 = low; 3 = medium; and 4 or 5 = high.

## 3. Results

### 3.1. Literature Selection

A total of 1284 articles were initially identified. Following review by title and abstract, 75 studies progressed to full manuscript review. Of these, 45 were excluded as they did not fulfill the inclusion criteria. The remaining 30 trials were included in this review. All 30 contributed to the qualitative analysis and 25 contributed to the quantitative analysis. The detailed process is illustrated in Figure 1.

### 3.2. Study Characteristics

Eighteen trials were conducted in upper middle-income countries [20,21,22,23,24,25,26,27,28,29,30,31,32,33,34,35,36,37,38], nine in lower middle-income countries [39,40,41,42,43,44,45,46,47] and two in low-income countries [48,49]. The total sample size was 16,670, with individual study participants ranging from 41 [49] to 8120 [40], with 11 of the 30 studies included having a sample size greater than 200. Mean age of participants was 55.4 (SD = 5.6) years and 58% were female (*n* = 9521). Diabetes duration ranged from 2.3 to 11 years, but was only reported in 12 studies, and diabetes medication was only reported in six studies. Study duration ranged from four weeks [28] to 348 weeks [35], with more than a third of studies exceeding a one-year duration. The most commonly investigated outcome measures were HBA1c (*n* = 20 studies), FBG (*n* = 13) and BMI (*n* = 13). Table 1 provides a detailed account of all study characteristics.

### 3.3. Lifestyle Intervention Characteristics

Of the 30 included trials, 24 of the interventions were described as diabetes self-management education programs [20,21,22,23,25,26,27,29,32,33,34,35,36,37,38,39,40,41,42,43,44,45,46,48], where the focus was on disease management skills including decision making, problem solving, and action planning. These programs were multifactorial in terms of their content. The majority included education relating to physical activity and diet (*n* = 19) and medication advice (*n* = 18). Other content covered was foot and skin care (*n* = 12), glucose monitoring (*n* = 14), smoking cessation (*n* = 7), stress control (*n* = 2) and blood pressure monitoring (*n* = 1). Of the remaining six trials, three investigated the effectiveness of nutrition/diet interventions [28,30,31], two prescribed exercise programs [47,49] and one was a combined diet and exercise program [24].

In 16 trials, the intervention was delivered by healthcare professionals [21,22,23,24,28,30,32,33,35,37,39,42,44,45,46,47]. Trained educators delivered five programs [23,36,41,43,48] and peer leaders or lay facilitators delivered three [26,38,40]. In one trial, the intervention was delivered by the researchers [25] and, in another, it was delivered online [31]. Five trials did not provide information relating to intervention delivery [20,27,29,34,49].

In terms of where the interventions were delivered, eighteen took place in either a hospital or healthcare setting [20,21,27,29,30,31,32,33,34,35,37,39,41,44,45,46,47,48] but did not differentiate between hospital and clinic settings. Five interventions were conducted in community health centers [24,26,36,38,40], three took place at home [22,40,43], two were reported as being in any location where technology was accessible [25,31] and the remaining two did not report a location for intervention delivery [23,49]. Appendix A provides a complete overview of the interventions and all their characteristics.

### 3.4. Effects of Lifestyle Interventions on Clinical Outcome Measures

A meta-analysis was conducted on three clinical outcomes—HbA1c, FBG and BMI, as illustrated in Figure 2. Twenty of the 30 included trials provided post-intervention data on HbA1c and data from 19 trials were included in the meta-analysis. One trial was excluded as it measured HbA1c at day 30.

Compared to the control group, in 19 trials with 5428 participants, intervention group participants reduced their HbA1c by 0.63% (95% CI, −0.86 to −0.40). Thirteen trials (*n* = 3306) were included in the meta-analysis exploring FBG, with results revealing a small reduction in FBG in the intervention group compared to control group (−0.35; 95% CI, −0.54 to −0.16). Finally, when compared to control group participants, intervention group participants reduced their BMI by 0.5 kg/m^2^ (95% CI −0.8 to −0.2) in 13 trials with 3461 participants. Details of all the other clinical outcomes reported in the studies can be found in Appendix A.

A subgroup analysis provided data relating to the effectiveness of the intervention type and its delivery. The self-management education programs that included multi-components demonstrated moderate to large effects on HbA1c, FBG and BMI, while interventions that focused specifically on structured diet/exercise or a combination of both demonstrated no effect or small effects (Table 2). Programs delivered by healthcare professionals demonstrated larger effects on HbA1c and FBG than those that were peer-led. Additionally, in terms of settings, interventions delivered in healthcare or/hospital settings appeared to be more effective at improving glycemic control than those delivered in the community or at home.

### 3.5. Risk of Bias Assessment

Twelve trials were rated as “weak”, 14 as “moderate,” and four as “strong” on the Effective Public Health Practice Project quality assessment tool. No trial was categorized as strong in all six components. Selection bias was apparent in most studies. Most authors did not report the rationale behind their participant selection. The randomization procedure was described adequately in 20 trials, with the remaining 10 using the word “random”; however, these trials did not describe a randomization process [21,22,23,31,37,44,45,46,48,49]. A major source of bias identified across all trials was blinding. Apart from three trials [30,36,39], it was not stated clearly whether the trial participants were aware of the research question. Data collection was recorded as strong, apart from in four studies; three were classified as weak [22] and one was classified as moderate [31,36,39]. Studies provided a range of detail on participant attrition, with only one providing complete numbers [21] and others not reporting drop-out rates at all [30,36,39]. Further details relating to trial quality are provided in Appendix A.

### 3.6. Publication Bias

As we included ≥10 studies in the meta-analysis of HbA1c, FBG and BMI, we generated funnel plots. The funnel plots were nearly symmetrical and every meta-analysis exhibited negative and positive results, which meant that there is little possibility of publication bias in this study. See Appendix A for detailed information.

### 3.7. Strength of the Evidence

There was high quality evidence that lifestyle intervention, using self-management education, resulted in a significant decrease in HbA1c and BMI, with moderate quality evidence for its effectiveness in decreasing FBG. The quality of the evidence in relation to the different subgroups was varied, but mostly deemed to be of moderate or low quality, with 50% (9/18 subgroups) categorized as low or very low. Most subgroups were downgraded due to a high risk of bias in most of the included trials, high heterogeneity between trials and imprecision of results. Appendix A provides details.

## 4. Discussion

This systematic review analyzed the evidence from 30 RCTs with 16,670 participants, to assess the effect of lifestyle interventions on the management of T2DM in LMICs. Overall, the results are positive, showing these interventions can improve key clinical markers in T2DM, including HbA1c, FBG and BMI. The primary outcome in this review was improvement in glycemic control (HbA1c) and a meta-analysis including 19 trials found a significant reduction in HbA1c (−0.63%; 95%CI, −0.86 to −0.40). Our subgroup analyses further explored core characteristics of included lifestyle interventions. Findings suggest: (1) self-management education programs may be more effective than lifestyle modification that focused specifically on diet and exercise, (2) interventions delivered by healthcare professionals (HCPs) resulted in better outcomes than those delivered by peers or lay educators, and (3) interventions delivered in the clinical environment were superior to those delivered in the community or home settings.

The reduction in HbA1 found in this study is clinically significant [50] and has been shown to have a positive effect on both morbidity and mortality. A systematic review of trials that decreased HbA1c by at least 0.5% found that they were associated with a significant reduction in incident cardiovascular events and myocardial infarction, commonly recognized complications of T2DM [51]. Furthermore, a recent update on the cost effectiveness of interventions to manage T2DM categorized diabetes self-management programs as very cost effective [52], suggesting they are a very worthwhile component of a multipronged approach in the management of T2DM.

With reference to previous research, our findings are both confirming and conflicting. From the perspective of program types, we found that participants in self-management education programs had better clinical outcomes than those availed by an intervention that focused solely on diet and/or exercise. This is contrary to the findings reported by Huang et al. [10], where physical activity and diet modifications yielded a greater reduction in HbA1c and BMI than education programs. Interestingly, though, while we found self-management education programs clinically effective in reducing HbA1 (−8.1% HbA1c), our improvements following diet/exercise modification (−0.3% HbA1c) cannot be classified as clinically effective (minimum change of −0.5% HbA1c) [50] and were less than those reported by Huang et al. (−0.30mmol).

Although the interventions in both our review and that of Huang et al. were categorized as self-management education or exercise/diet modification, there was still significant heterogeneity between programs. For example, Huang et al. only classified a program as an exercise intervention if it was individualized and strictly supervised. Indeed, supervised exercise programs may be superior to unsupervised exercise for glycemic control and weight loss in people with T2DM [53]. Furthermore, they reported exercise and diet as two separate programs. In this current review, we classified an intervention as an exercise intervention if its primary focus was on physical activity or exercise and we did not stipulate supervision or individual programming as inclusion criteria. We also combined exercise and/or diet interventions to conduct the analysis.

A number of other reviews have been published that confirm our findings, illustrating that, irrespective of type, interventions focused on lifestyle change, are effective for the management of T2DM [11,12,54]. When compared to a control group, these reviews, exploring self-management education [11] in the Asian Western Pacific region, diet/exercise modification [12] in a Southeast Asian population and a combination of both (primarily in high-income countries) [54], reported a significant reduction in HbA1c and BMI from the baseline in the lifestyle intervention groups. Specifically with regard to LMICs, a very recent review exploring health system interventions for adults with T2DM with similar cohort selection criteria to ours found diabetes multicomponent self-management education interventions to be successful at improving glycemic control [55].

In clinical practice, intervention setting and by whom the intervention is delivered have been shown to be important influencers of outcomes due to their potential to influence uptake and attrition [56,57]. Our sub-group meta-analyses was designed to explicitly explore the effect of these factors which, to the best of our knowledge, has not previously been conducted in reviews of T2DM lifestyle management in LMICs. Our findings showed that programs delivered by HCPs in a clinical environment were more effective in reducing HbA1c than those delivered by peer educators at home or in a community setting. Previous reviews have reported on the differences between interventions that are delivered by one healthcare professional versus a multidisciplinary team [11,55], showing a team approach to be more effective. A very recent review exploring the effectiveness of peer- and community health worker-led self-management education programs in LMICs showed inconsistent improvements in clinical, behavior and psychological outcomes [58]. Ideally, based on the findings of our current review alongside those previously reported, T2DM lifestyle interventions, be they self-management education or active lifestyle modification using diet and exercise, would be led by HCPs and delivered in a clinical setting.

However, this is perhaps an unrealistic goal, as most LMICs do not have the extensive health systems of high-income countries nor the number of HCPs working in the field of chronic disease management [59]. An alternative may be the use of healthcare–community partnerships. This approach differs from a peer-led process as it ensures the patient has the opportunity to interact with an HCP for initial education/goal setting and again at a number of agreed time points over the intervention duration. In between, patients return to their local communities, where a support mechanism using peer led educators is in place to facilitate and sustain healthy behaviors [60]. This may be a viable option as, while it still includes HCP input, it is less demanding on healthcare personnel and has the potential to reach a much larger number of patients. It has been used successfully in HICs but further research into its feasibility and efficacy in LMICs is required.

This study has several strengths and weaknesses. The review was systematic and exhaustive. A considerable sample size of adults living with T2DM (*n* = 16,670) was included, thus providing us with the power to detect statistically significant mean differences in glycemic control. Only RCTs, the gold standard for evaluating the effectiveness of an intervention, were included. Our review shares a number of limitations with the studies on which it is based. Although we attempted to limit heterogeneity by using strict inclusion and exclusion criteria, we fully acknowledge that the lifestyle intervention are diverse and, although data were analyzed according to intervention sub-type and delivery, the wide variability in terms of age, sex, and duration of follow-up could have confounded our findings. Due to the observed heterogeneity in terms of intervention duration, we conducted some pre-liminary exploration but found it to be meaningless because of the frequency versus duration issues, i.e., some intense studies may have been short in duration and some studies with a really long follow-up may have only had two hours of intervention. Furthermore, defining lifestyle intervention is challenging and there is no available consensus definition. In terms of outcomes, this review mainly focused on glycemic control (HbA1c and FBG). This was due to the limited availability of secondary clinical outcomes data (other metabolic markers) and the issue of calculating an effect size for outcomes that were reported using several different measurement tools (e.g., quality of life, self-efficacy, diabetes knowledge). Finally, 26 of the included RCTs were recorded as weak or moderate on the Effective Public Health Practice Project quality assessment tool [18]; hence, the results of this present review should be interpreted in a conservative manner.

Future research should consider some of the limitations noted above. Gaining consensus around the definition of a lifestyle intervention, maybe specifically in relation to LMIC, would be useful and could perhaps address some of the homogeneity issues we came across when analyzing the data in this review; for example, in duration to intervention duration and content. In terms of outcomes that should be measured, a recent consensus study identified 18 core outcomes across five domains for randomized effectiveness trials in T2DM [61]. The implementation of such a core outcome set in future trials could help to ensure that outcomes of importance to all stakeholders are measured and reported. This, in turn, will facilitate the comparison of results across trials, allow for pooled analyses in reviews and, inevitably, result in enhanced relevance of reported research findings. Finally, based on the GRADE quality of evidence assessment, the evidence available for lifestyle interventions delivered in the home setting was deemed to be of a low quality. This was, in part, due to a lack of studies conducted in this environment, highlighting the necessity for further exploration.

## 5. Conclusions

Despite its limitations, this review identified and summarized the available evidence from LMICs regarding the effectiveness of lifestyle interventions in the management of adults living with T2DM. The results suggest that overall lifestyle interventions are associated with clinically significant improvements in glycemic control. Interventions that promote self-management, including multiple education components, delivered by healthcare professionals in a clinical setting appear to be optimal. However, this is perhaps unworkable in terms of population reach in LMICs, due to a shortage of resources and HCPs working in the area of chronic disease management. Exploring the option of a hybrid model using health–community and home delivery might be a more realistic option. There is, however, a need to further investigate such options in well-designed trials to better inform this evidence base.

## Figures and Tables

**Figure 1 ijerph-18-06273-f001:**
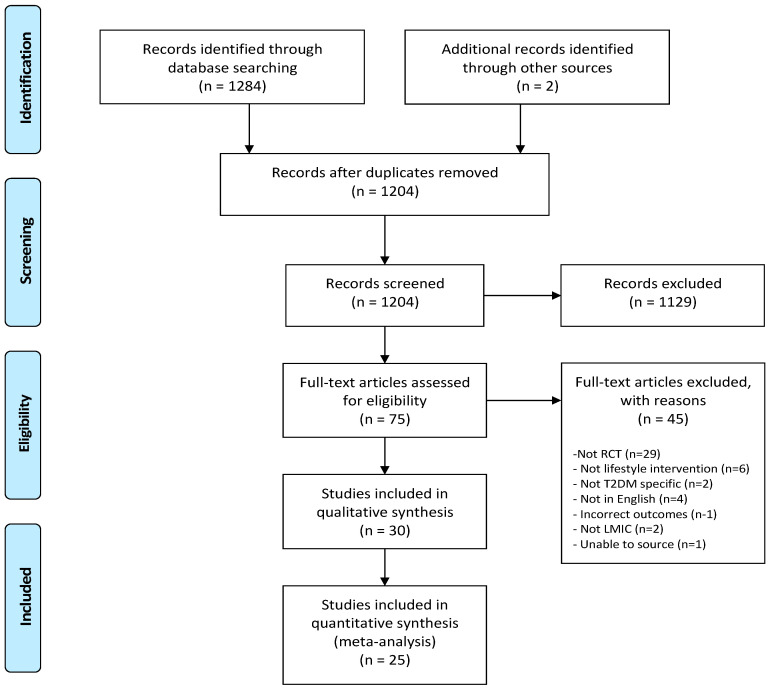
PRISMA study flow diagram.

**Figure 2 ijerph-18-06273-f002:**
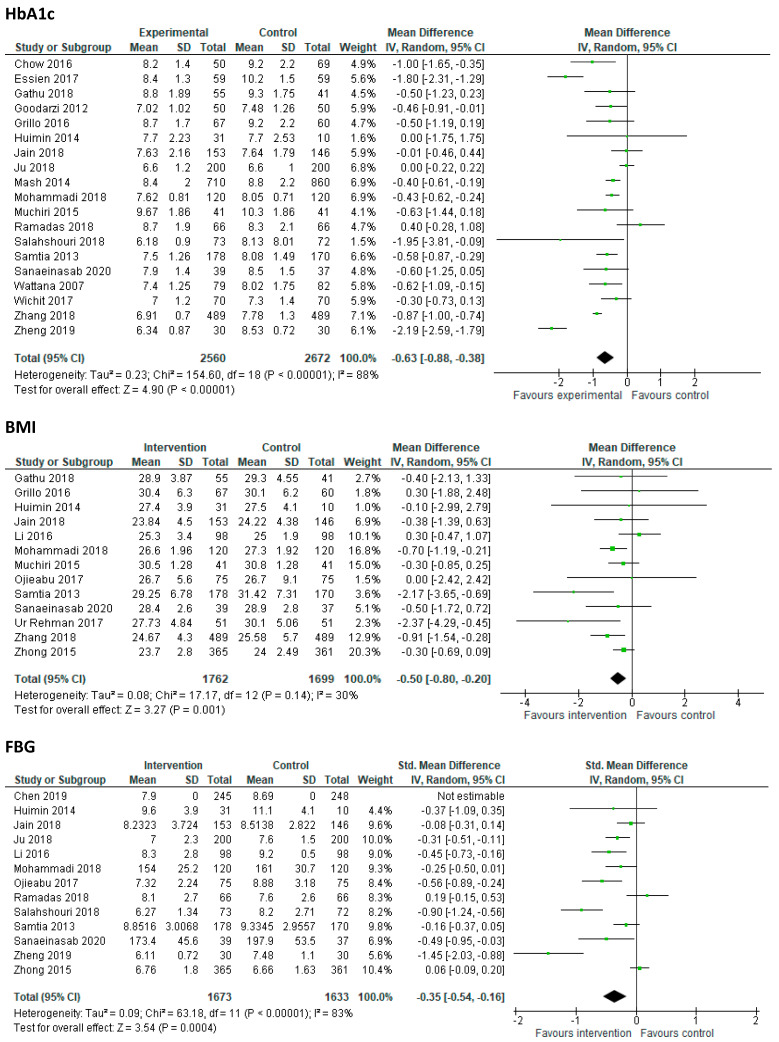
Forest plots illustrating group data from meta-analysis of HbA1c, FBG and BMI.

**Table 1 ijerph-18-06273-t001:** Characteristics of studies included in the review.

Author	Year	Country	Country Classification	Intervention Description	Intervnetion Category	Study Duration (Weeks)	Sample Size (I/C)	Age Mean (SD)	Gender (% Female)	Diabetes Any Diabetes Medication (Insulin)	Outcome Measures
Chaveepojnkamjorn [20]	2009	Thailand	Upper middle	Self-help group program for patients.	SME	16	80/84	I: 48.9 (6.9) C: 49.1 (7.3)	I: 78 C: 76		QoL
Chen [21]	2019	China	Upper middle	Education-based intervention for patients.	SME	104	245/248	I: 63.3 (6.8) C: 61.5 (9.1)	I: 79 C: 67		FBG, QoL
Chow [22]	2016	Malaysia	Upper middle	Home based pharmacist led educational intervention for patients.	SME	12	75/75	60.31	I: 64 C: 63		HbA1c
Debussche [48]	2018	Mali	Low	Structured T2DM Self-management Education by peers for patients.	SME	52	76/75	I: 53.9 (9.8) C: 51.1 (9.6)	I: 75 C: 77	I: 55 (22) C: 53 (21)	HbA1c, BMI, WC, SBP, DBP
Essien [39]	2017	Nigeria	Lower middle	Intensive diabetes self-management program for patients.	SME	26	59/59	I: 52.6 (10.9) C: 52.8 (10.1)	I: 52 C: 67		HbA1c
Fottrell [40]	2019	Bangladesh	Lower middle	1. MHealth educational voice messages. 2. Participatory learning which engages communities to identify and address their own local problems.	1. SME * 2. SME	104	* 4093/4079/4108		I 1: 55 I 2: 53 C: 52		BMI, SBP, DBP, QoL
Gagliardino [23]	2013	Argentina	Upper middle	1. Education for healthcare professionals. 2. Self-management education for patients. 3. Education for patients and healthcare professionals.	1. SME ^ 2. SME ^^ 3. SME ^^^	182	135/135/135/135	I 1: 62.4 (9.1) I 2: 62.2 (8.4) I 3: 62.2 (9.0) C: 62.1 (8.4)			HbA1c, SBP, TG
Gathu [41]	2018	Kenya	Lower middle	Diabetes self-management education training for patients.	SME	26	70/70	I: 50.2 (9.9) C: 47.5 (9.5)	I: 41 C: 47		HbA1c, BMI, SBP, DBP
Goldhaber-Fiebert [24]	2003	Costa Rica	Upper middle	Community-based nutrition and exercise intervention for patients.	SDEP	12	40/35	I: 60 (10) C: 57 (9)	I: 83 C: 75		HbA1c, FBG, BW, BMI, SBP, DBP, TC, HDLc, LDLc, TG
Goodarzi [25]	2012	Iran	Upper middle	Distance education including exercise, diet, diabetic meds, self-monitoring of blood glucose levels, via mobile phone text messaging.	SME	12	50/50	I: 50.1 (10.3) C: 56.7 (9.7)	I: 79 C: 76	I: 41 (13) C: 37 (13)	HBA1c, TC, HDLc, LDLc, TG
Grillo [42]	2016	Brazil	Lower middle	Structured diabetes self-management education course for patients, administered by a generalist nurse trained in diabetes education.	SME	52	69/68	I: 61.7 (9.9) C: 63.2 (9.7)	I: 71 C: 55	I: 69 (4) C: 68 (3)	BMI, WC, SBP, DPB, HDLc, LDLc
Huimin [49]	2014	Mozmbique	Low	Low to vigorous intensity exercise intervention for patients.	SDEP	12	31/10	I: 53.2 (2.5) C: 55.3 (3)	I: 0 C: 0		HbA1c, FBG, BW, BMI, WC, SBP, DBP, VO2max
Jain [43]	2018	India	Lower middle	Face-to-face interaction by community health workers as well as telephonic reminders.	SME	24	153/146	I: 55.7 (10.9) C: 57.4 (10.1)	I: 45 C: 41		HBA1c, FBG, BW, BMI, WC, SBP, DPB, TC, HDLc, LDLc, TG
Ju [26]	2018	China	Upper middle	Usual education of 2 h each month of focused diabetes education and peer support.	SME	52	200/200	I: 67.8 (7.4) C: 68.8 (8)	I: 65 C: 68		HBA1c, FBG, 2HBG
Li [28]	2016	China	Upper middle	Intensive nutrition education classes for patients.	SDEP	4	98/98	I: 59.1 (4.6) C: 58.3 (4.1)	I: 52 C: 46	I: 84 (16) C: 82 (17)	HBA1c, FBG, 2HBG, BW, BMI, HDLc, LDLc, TG
Malathy [44]	2011	India	Lower middle	Pharmacist-led diabetes counseling program for patients.	SME	12	137/70	I: 52.1 (9.5) C: 51.1 (9.8)	I: 73 C: 35		TC, HDLc, LDLc, TG
Mash [36]	2014	South Africa	Upper middle	Group-based diabetes education program for patients.	SME	52	710/860	I: 55.8 (11.5) C: 56.4 (11.6)	I: 72 C: 76	I: 213 (59) C: 228 (73)	HBA1c, WC, BW, SBP, DBP, TC
Mohammadi [29]	2018	Iran	Upper middle	Self-efficacy education for patients, based on health belief model.	SME	36	120/120	I: 51.2 (6.2) C: 51.4 (6.1)			HBA1c, FBG, BW, BMI, WC, LDLc, HDLc, QoL
Muchiri [30]	2015	South Africa	Upper middle	Nutrition education program for patients.	SDEP	52	41/41	I: 59.4 (6.9) C: 58.2 (8)	85		HbA1c, BMI, SBP, DBP, TC, LDLc, HDLc
Ojieabu [45]	2017	Nigeria	Lower middle	Pharmacist-led education and counseling program for patients.	SME	20	75/75		I: 64 C: 60		FBG, BMI, SBP, DBP
Ramadas [31]	2018	Malaysia	Upper middle	Web-based dietry intervention program.	SDEP	52	66/66	I: 49.6 (10.7) C: 51.5 (10.3)	I: 41 C:40	I: 56 (7) C: 51 (7)	HbA1c
Salahshouri [32]	2018	Iran	Upper middle	Educational sessions for patients, based on psychological factors administered by a group of internal specialists, dieticians, diabetes experts, a psychologist and a religious expert.	SME	20	73/72	I: 55.9 (12.4) C: 54.8 (9.4)	I: 67 C: 69		HbA1c, FBG
Sanaeinasab [27]	2020	Iran	Upper middle	Group face-to-face sessions, 1 × weekly × 6 weeks; health education and promotion program focusing on diabetes self-care.	SME	6	40/40	50.7 (5.9)	I: 55 C: 63		HBA1c, FBG, BMI, SBP, DBP, TC, HDL, LDL, TG
Samtia [46]	2013	Pakistan	Lower middle	Pre-defined, pharmacist-led, multifactorial, specialized care for patients.	SME	20	178/170	I: 46.1 C: 42.3	I: 47 C: 52		HBA1c, FBG, BMI, WC
Ur Rehman [47]	2017	Pakistan	Lower middle	Supervised structured aerobic exercise training (SSAET) program, routine medication, and dietry planning for patients.	SDEP	25	51/51	54.7 (8.2)	67		BMI
Wattana [37]	2007	Thailand	Upper middle	Diabetes self-management program for patients, based on the theories of self efficacy and self-management.	SME	24	79/82	56.8 (4.6)	76		HbA1c, QoL
Wichit [33]	2017	Thailand	Upper middle	Family oriented self-management intervention program for patients, designed based on the self-efficacy theory.	SME	13	70/70	I: 61.3 (11.6) C: 55.5 (10.5)	I: 76 C: 70		HbA1c, QoL
Zhang [35]	2018	China	Upper middle	A nine-component systematic education intervention for patients.	SME	348	489/489	I: 56.8 (14.2) C: 52.6 (13.2)	I: 51 C: 49		HbA1c, BMI, SBP, DBP, TC, LDLc, HDLc
Zheng [34]	2019	China	Upper middle	2-session diabetes self-management education program for patients with theory and practical elements.	SME	12	30/30	I: 52.5 (10.5) C: 51.9 (12.3)	I: 47 C: 43		HBA1c, FBG, 2HBG
Zhong [38]	2015	China	Upper middle	Peer leader-support program for patients.	SME	26	365/361		55		FBG, 2HBG, BMI, SBP, DBP

I: Intervention. SDEP (Structured diet and/or exercise program): A structured nutrition/exercise/combined intervention, where the focus was on a prescribed and/or supervised individualized program. SME (Self-management education) consists of a multifactorial program focusing on education provision education around multiple risk factors associated with T2DM. C: Control. * mHealth self management education. ^ Self management education delivered to patients, ^^ SME delivered to healthcare practitioners, ^^^ SME delivered to patients and healthcare practitioners. QoL—Quality of life, CV—cardiovascular, BP—blood Pressure, HbA1c—glycated hemoglobin (%), FBG—fasting blood glucose (mmol/L), FBS—fasting blood glucose, BMI—body mass index (kg/m^2^), WC—waist circumference (cm), SBP—systolic blood pressure (mmHg), DBP—diastolic blood pressure (mmHg), TG—triglycerides (mmol/L), TC—total cholesterol (mmol/L), HDLc—high density lipid cholesterol (mmol/L), LDLc—low density lipid cholesterol (mmol/L), 2HBG—2 h blood glucose (mmol/L), BW—body weight (kg).

**Table 2 ijerph-18-06273-t002:** Subgroup analysis, based on intervention type, delivery and setting.

	Clinical Outcome Measure		
	Glycated Hemoglobin (HbA1c)	Body Mass Index (BMI)	Fasting Blood Glucose (FBG)
Overall	MD −0.63 (−0.86 to −0.40), I^2^ 88%	MD −0.50 (−0.80 to −0.20), I^2^ 30%	SMD −0.35 (−0.54 to −0.16), I^2^ 83%
Intervention			
*Self-management education*	MD −0.69 (−0.96 to −0.43), I^2^ 90%	MD −0.54 (−0.79 to −0.29), I^2^ 0%	SMD −0.42 (−0.69 to −0.16), I^2^ 87%
*Structured Diet/exercise/combined*	MD −0.07 (−0.81 to 0.67), I^2^ 45%	MD −0.06 (−0.64 to 0.51), I^2^ 35%	SMD −0.14 (−0.76 to 0.48), I^2^ 87%
Delivery			
*Healthcare professional(s)/MDT*	MD −0.71 (−1.01 to −0.41), I^2^ 77%	MD −0.48 (−1.11 to 0.15), I^2^ 57%	SMD −0.37 (−0.70 to −0.05), I^2^ 84%
*Trained peers/lay people*	MD −0.24 (−0.47 to 0.00), I^2^ 57%	MD −0.51 (−1.10 to 0.09), I^2^ 30%	SMD −0.11 (−0.33 to 0.12), I^2^ 77%
Setting			
*Hospital/clinic*	MD −0.77 (−1.08 to −0.56), I^2^ 88%	MD −0.61 (−1.07 to −0.15), I^2^ 50%	SMD −0.47 (−0.74 to −0.20), I^2^ 82%
*Community*	MD −0.2 (−0.59 to 0.19), I^2^ 85%	N/A	SMD −0.12 (−0.48 to 0.24), I^2^ 88%
*Home*	MD −0.48 (−1.44 to 0.49), I^2^ 83%	N/A	N/A

MD = Mean difference; SMD = standardised mean difference; N/A = not applicable, as ≤1 study in analysis.

## Data Availability

All data extracted and analyzed are contained within the article or Appendix A.

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
