# Peer review of "Lifestyle Interventions to Improve Glycemic Control in Adults with Type 2 Diabetes Living in Low-and-Middle Income Countries: A Systematic Review and Meta-Analysis of Randomized Controlled Trials (RCTs)"

_ijerph, 2021, doi:10.3390/ijerph18126273_

Round 1

Reviewer 1 Report

  1. Since the primary efficacy outcome was change in HbA1c, it sounds reasonable that the authors should restrict their analysis to studies with a duration greater than 12 weeks. Please update, by providing corresponding analyses.
  2.  Please provide grading of evidence according to GRADE.
  3. Did the authors evaluate selected RCTs for publication bias?
  4. Data regarding the baseline antidiabetic medication across the selected RCTs is missing. Please provide if available.
  5. PRISMA checklist is missing.

Author Response

Please find attached a pdf document that provides a point-by-point response to the review report 

Reviewer 2 Report

The review article entitled  "Lifestyle interventions to improve glycemic control in adults 2 with type 2 diabetes living in Low-and-Middle Income Countries: A Systematic Review and Meta-analysis of Randomized 4 Controlled Trials (RCTs)" by O’Donoghue et al. have summarized the available evidence from low-and-middle income countries regarding the effectiveness of lifestyle interventions in the management of adults living with type 2 diabetes. This review article has been arranged in such a way that it looks like resemble to the currently published review article entitled “Education‑based, lifestyle intervention programs with unsupervised exercise improve outcomes in adults with metabolic syndrome. A systematic review and meta‑analysis” in Reviews in Endocrine and Metabolic Disorders" by same authors. The mentioned review article has been published on March 8, 2021 as compared to currently submitted review article on April, 2021. This review almost replicates of the published one recently. I think that this article will go against copyright rule of the current journal. I can't recommend this review to be published in the 'International Journal of Environmental Research and Public Health'

Author Response

Please find attached a PDF doc that provides point-by-point responses to the review report 

Reviewer 3 Report

Dear authors,

The study is quite interesting and easy to read and to understand. I have some minor comments:

  • The link to the supplementary material does not work. I have been seen the material and it has a great quality and great information for the readers. Thus, please be sure that the link works so the readers can read it.
  • Is it possible to do a meta-analysis controlling the variable "intervention duration"?

Kind regards

Author Response

(The authors gave the same response as above.)

Reviewer 4 Report

See attached file

Author Response

(The authors gave the same response as above.)

Round 2

Reviewer 4 Report

I appreciate the authors responses and edits. See below for remaining minor edits:

  1. Line 39: add comma after “(T2DM)”
  2. Line 50: add “improving” before “glucose control”
  3. Line 56: Add comma after “diabetes self-management education [11]”
  4. Line 58: remove comma after “LMICs”
  5. Line 166: I believe “week” should be “weak”
  6. Line 167: remove “do” before “the quantitative analysis”
  7. Line 325: change “was” to “were” (plots were nearly symmetrical)
  8. Line 326: change “exited” to “exhibited”
  9. Line 335: Define “ROB”
  10. Line 371: add period after “et al”
  11. Line 411: Define “HIC”
  12. Line 419: can a reference be added to support that this described method has been successfully used in HICs?
  13. Line 446: add comma after “LMIC”

Author Response

Attached an updated manuscript including all the grammatical changes requested by reviewer 4. 
